# Ongoing Failure to Deliver Guideline-Concordant Care for Patients with Pancreatic Cancer

**DOI:** 10.3390/cancers17020170

**Published:** 2025-01-07

**Authors:** Jonathan Ejie, Amir Ashraf Ganjouei, Sophia Hernandez, Jaeyun Jane Wang, Fernanda Romero-Hernandez, Laleh Foroutani, Kenzo Hirose, Eric Nakakura, Carlos Uriel Corvera, Adnan Alseidi, Mohamed Abdelgadir Adam

**Affiliations:** 1Department of Surgery, University of California San Francisco, San Francisco, CA 94143, USA; jonathan.ejie@ucsf.edu (J.E.); sophia.hernandez@ucsf.edu (S.H.); jaeyunjane.wang@ucsf.edu (J.J.W.); maria.romerohernandez@ucsf.edu (F.R.-H.); laleh.foroutani@ucsf.edu (L.F.);; 2School of Medicine, University of California San Francisco, San Francisco, CA 94143, USA

**Keywords:** guideline-concordant care, pancreatic cancer, pancreatic neoplasm, surgery, pancreatectomy

## Abstract

Pancreatic ductal adenocarcinoma (PDAC) is one of the deadliest cancers, with only 13% of patients surviving beyond five years. Effective treatment often requires a combination of surgery, chemotherapy, and/or radiation, depending on the stage of the disease. However, many patients do not receive care that follows established guidelines. In this study, we analyzed the treatment patterns of over 50,000 patients in California and found that fewer than half received guideline-concordant care. Patients who did receive care aligned with guidelines lived significantly longer, with a median survival of 10 months compared to 3 months for others. Certain groups, including Hispanic and Black patients, those without insurance, and those with other health conditions, were less likely to receive proper care. These findings underscore the need to address barriers to guideline-based treatment to improve survival rates and outcomes for patients with PDAC.

## 1. Introduction

Pancreatic ductal adenocarcinoma (PDAC) is one of the deadliest cancers in the United States, with a 5-year survival rate of only 13% [1]. It is projected to become the third and fourth leading cause of cancer deaths in women and men, respectively, by 2024 [2,3]. Most patients are diagnosed with locally advanced (30–35%) or metastatic disease (50–55%) [4]. While there have been advancements in the treatment options and survival rates of other gastrointestinal cancers, such developments in PDAC remain relatively stagnant [3,5]. Central to improving survival outcomes for PDAC is the delivery of multidisciplinary, stage-specific care, as outlined in clinical guidelines [6].

Guideline-concordant care (GCC), defined as adherence to evidence-based practices and standardized treatment protocols, has increased overall survival in patients with PDAC [7,8]. The National Comprehensive Cancer Network (NCCN) is an alliance of leading cancer centers dedicated to improving and standardizing cancer care globally. For early-stage PDAC, the NCCN guidelines recommend surgery with chemotherapy or radiation as the primary treatment approach. In contrast, for advanced-stage disease (stages 3 and 4), chemotherapy/radiation is the cornerstone of management. Yet, previous studies suggest that as little as 43% of patients receive GCC [7]. Providing GCC can be challenging in PDAC, as treatment approaches vary widely based on the stage of the disease. As the cancer progresses, the treatment strategies become less standardized [9]. This and institutional treatment variations can make it difficult to ensure uniform adherence to guidelines.

Although adherence to GCC improves overall survival, limited data exist based on stage-specific recommendations and within a diverse patient population. Therefore, this paper aims to examine the adherence of GCC in patients with PDAC within the State of California, a densely populated and diverse state in the USA [10]. Specifically, we are interested in assessing factors associated with receiving GCC, its association with overall survival, and understanding variations in adherence across different cancer stages. By identifying areas where care diverges from established guidelines, this study sought to reveal opportunities for enhancing treatment consistency and improving patient outcomes in this challenging malignancy.

## 2. Materials and Methods

### 2.1. Data Source and Study Population

The primary data source for this study was the California Cancer Registry (CCR) [11]. This statewide, population-based cancer surveillance system collects comprehensive information on all new cancer diagnoses and related mortality within California. We included patients diagnosed with PDAC from 2004 to 2020. Eligibility was determined using specific histology codes based on the International Classification of Diseases for Oncology, Third Edition (ICD-O-3) [12].

Selected variables for analysis in this study encompassed demographic factors and a range of clinical features, including cancer location, primary surgical site, cancer stage, and the Charlson comorbidity index. Treatment-related variables, such as chemotherapy and radiation therapy, and reasons for not administering treatment were also examined.

### 2.2. GCC Definition

GCC was determined according to the National Comprehensive Cancer Network (NCCN) guidelines [13]. For patients diagnosed with stage 1 or 2 pancreatic cancer, care was considered guideline-concordant if it included pancreatectomy supplemented by chemotherapy or chemoradiation. For patients with stage 3 disease, GCC was defined as receiving chemotherapy, with or without accompanying pancreatectomy. For those diagnosed with stage 4 pancreatic cancer, GCC was identified as chemotherapy alone [13]. Although the NCCN guidelines have evolved over time to incorporate advancements in research and clinical practice, the general stage-based approach to the management of PDAC has remained consistent throughout the duration of our study period [6,14,15].

### 2.3. Statistical Analyses

Descriptive statistics were initially calculated to characterize the demographic and clinical features of the study population. A multivariable logistic regression model was employed to assess the relationship between patient characteristics and adherence to GCC. Survival outcomes for each disease stage group were then assessed using Kaplan–Meier survival curves. Lastly, a Cox proportional hazards model was applied to evaluate the hazards associated with various factors, such as age, sex, race, insurance status, rural-urban classification, comorbidity score, disease stage, and the receipt of guideline-concordant therapy. *p*-values less than 0.05 were considered significant. All statistical analyses were performed using R software, version 4.4.1 [16].

## 3. Results

### 3.1. Patient Characteristics

The study cohort included 50,346 patients diagnosed with PDAC in California from 2004 to 2020. The median age at diagnosis was 70 years, and females accounted for 48% of the cohort. Concerning ethnicities, 60% were White, 19% Hispanic, 12% Asian, and 7% Black. Most patients were insured, with 37% having private insurance and 57% having Medicare. Geographically, the vast majority (94%) resided in metropolitan areas, with a small percentage in micropolitan areas (4.2%). The distribution of cancer stages revealed that 54% of the patients were diagnosed at stage 4, whereas only 9.9% were diagnosed at stage 1. Detailed demographic and clinical characteristics of the cohort are presented in Table 1.

### 3.2. GCC Rates

The overall rate of GCC within our cohort was 46.7%. Concordance rates varied significantly by tumor stage, being lowest for stage 1 patients at 19.6%, increasing to 39.7% for stage 2, 50.4% for stage 4, and reaching the highest at 69% for stage 3 (Figure 1). Over the study period (2004–2020), stage 2 patients exhibited the most substantial increase in GCC, rising from 29% to 49%, while stage 1 saw the smallest increase, from 13.0% to 19.1% (Figure 2).

### 3.3. Factors Associated with GCC

Multivariable analysis indicated that patients with stage 2, 3, or 4 disease were significantly more likely to receive GCC compared to those diagnosed at stage 1 (Table 2). Racial disparities were evident, as Black and Hispanic patients were less likely to receive such care compared to White patients [odds ratios (OR) 0.74 (95% CI 0.68–0.80); *p* < 0.001] and [OR 0.78 (0.74–0.82); *p* < 0.001, respectively]. Insurance status is also associated with care concordance; patients with no insurance or only Medicare coverage were less likely to receive GCC compared to those with private insurance [OR 0.40 (0.34–0.47); *p* < 0.001] and [OR 0.95 (0.91–1.00); *p* = 0.042, respectively]. Additionally, patients with a Charlson comorbidity score of 2 or higher were less likely to receive GCC [OR 0.68 (0.65–0.72); *p* < 0.001]. Lastly, patients who were treated at a National Cancer Institute-designated cancer center were more likely to receive GCC than those who did not [OR 1.64 (1.56–1.72); *p* < 0.001] (Table 2).

### 3.4. Guideline Discordance in Stage 1 PDAC

Our analysis revealed that patients with stage 1 pancreatic cancer had the lowest rates of adherence to guideline-recommended care, at only 19.6%. Of the 4968 individuals diagnosed with stage 1 pancreatic adenocarcinoma, only 1542 (31%) underwent documented surgical intervention. Conversely, 2587 (52%) patients received some form of systemic therapy, primarily in the form of radiation. Additionally, the majority, 2381 (48%) patients, did not receive any systemic therapy, while 2862 patients lacked a documented reason for not having surgery as part of their initial treatment plan. Further examination of the CCR coding algorithm suggests that potential reasons for not undergoing surgery may have included patient acceptance of the “no treatment” option, where surgery was one of the multiple options, and the patient chose an alternative or the cancellation of surgery due to a complete response to radiation and/or systemic therapy.

### 3.5. Reasons for No Surgery for Stage 1

Among the 4966 patients with stage 1 PDAC, 3423 (69%) did not undergo surgery (Figure 3). Of these, the treatment plan of 2862 patients (84%) did not include surgery as part of their treatment plan. Other reasons for not receiving surgery included 4.1% due to contraindications, 4.3% who refused the recommended surgery, 0.4% who died prior to the planned surgery, 0.3% for whom surgery was recommended but not performed, and in 2% of the cases, it was unknown whether the surgery was performed. 

### 3.6. Survival Analysis

The overall median survival for patients receiving GCC was 10 months, compared to only 3 months for those not compliant with guidelines (Appendix A. More specifically, patients with stage 1 PDAC who received GCC had longer median survival than those who did not receive GCC (47 months vs. 8 months). Significant survival differences were also observed across different stages (Figure 4). After adjustment for confounders, patients who received GCC had significantly improved survival when compared to those who did not receive GCC [Hazard Ratio (HR) 0.39 (95% CI 0.38–0.40); *p* < 0.001)] (Table 3). Of note, patients who underwent surgery experienced a greater reduction in mortality risk compared to those receiving other therapies, regardless of guideline concordance (Appendix A). Black patients with PDAC had a statistically significant 4% [HR 1.04 (95% CI 1.00–1.08); *p* = 0.035)] compromise in survival compared to White patients, while Asian patients had a statistically significant 6% [HR 0.94 (95% CI 0.91–0.97); *p* < 0.001)] improvement in survival compared to White patients (Table 3). Furthermore, higher PDAC stages significantly increased mortality risks: patients with stage II, III, and IV disease had a 33%, 141%, and 365% increase in mortality compared to stage I, respectively. Additionally, a Charlson comorbidity score greater than two was associated with a 33% increase in mortality risk (Table 3).

These Kaplan–Meier curves depict the survival probabilities for pancreatic cancer patients at each disease stage (I–IV), comparing those who received GCC (red lines) against those who did not (blue lines). Across all stages, adherence to standard-of-care treatment is associated with significantly better survival outcomes, with a *p*-value of <0.0001.

## 4. Discussion

This population-based study underscores a critical gap in the delivery of guideline-adherent care for patients with pancreatic ductal adenocarcinoma in one of the largest and most resource-rich states in the USA. Despite the well-documented benefits of stage-specific, multidisciplinary treatment, only 47% of patients in our study received GCC. Notably, patients with stage 1 PDAC had the lowest rate of guideline adherence, primarily due to healthcare providers’ failure to recommend surgery during the early stages of their treatment plan. This finding is concerning, as adherence to GCC was associated with a substantial 61% improvement in survival benefit in our analysis. Moreover, among the four disease stages examined, patients with stage 1 PDAC who received GCC experienced the most significant survival benefits, with a median survival of 47 months compared to just 8 months for patients not receiving GCC. Importantly, our study also identified that patient’s race, age, and socioeconomic status significantly influenced the likelihood of receiving GCC.

Our analysis demonstrated that GCC was substantially lower for stage 1 pancreatic ductal adenocarcinoma patients compared to the overall patient population. This finding aligns with previous research, such as the studies by Hamad et al. and Tahome et al., who reported GCC rates of 14.6% and 19%, respectively, among stage 1 PDAC patients [17,18]. However, our study indicates an even lower surgical rate of 31% for stage 1 PDAC patients, contrasting with the 43.5% and 41% surgical rates observed in those earlier studies. The discrepancy may be explained by a greater emphasis on systemic therapy in our patient population, with a chemotherapy rate of 50% compared to the 33% and 43% rates reported in the previous studies (Appendix A). Similarly, the radiation therapy rate in our cohort was 10%, which contrasts with the higher rates of 18% and 27% observed in the earlier investigations (Appendix A). This is consistent with the regional disparities that exist in the treatment of stage 1 [19,20]. However, the results should be interpreted with caution due to the different sources of data (CCR vs. NCDB) and time periods that may influence the management approach between study populations. Future studies should focus on exploring the obstacles that limit the concurrent administration of systemic therapy and surgical resection at high frequencies. Understanding and addressing these barriers is crucial to improving the delivery of comprehensive GCC for pancreatic cancer patients.

We found that guideline adherence for patients with stage 1 pancreatic ductal adenocarcinoma has not seen the same level of improvement observed in other disease stages over the 2004 to 2020 period. This aligns with the findings of Hamad et al., who noted a concerning decline in stage 1 PDAC GCC compliance from 19% to 12%, even as GCC increased for stage 2–4 PDAC during the same timeframe [17]. Conversely, Dimou et al.’s 2004–2011 study reported an increase in multimodal therapy utilization for both stage 1 and 2 PDAC, largely driven by a rise in neoadjuvant therapy [21]. However, the grouping of stages 1 and 2 together in that analysis may have obscured the nuanced differences in GCC trends between the early and more advanced disease stages. Previous research suggests that patients with advanced-stage pancreatic ductal adenocarcinoma benefit from accessing specialized treatment centers to receive GCC [9,22,23]. Additionally, it is important to investigate whether persistent providers’ nihilistic attitudes towards PDAC may also contribute to the lack of treatments [24,25,26]. For instance, a study in Canada found that many physicians overestimated the mortality rate of pancreatic surgery and doubted whether surgery could cure pancreatic cancer [21].

Our study found that socioeconomic factors contributed to the care gap. Black and Hispanic patients were as likely as White patients to refuse surgery, but Black patients were less likely to have surgery as part of their initial treatment plan (Appendix A). As a result, Black patients had lower overall survival compared to White patients. These racial and socioeconomic disparities mirror those seen in other cancers and in patients with PDAC [27,28]. Fromer et al. found that among patients with stage 1 and 2 PDAC who undergo neoadjuvant chemotherapy nationally, Black and Hispanic patients were less likely to undergo subsequent surgery [10]. Hamad et al. also evaluated stage-specific GCC in patients with PDAC and found that older age, Black race, and increased Charlson comorbidity score were associated with decreased odds of obtaining GCC [17]. However, both cohorts come from NCDB and were >80% White and, therefore, less generalizable. Therefore, our study confirms racial–socioeconomic disparities in accessing GCC in a more diverse patient population in California. While this contributes to the validity and generalizability of this study, it also suggests that many patients throughout the United States may not be accessing optimal care. To address these disparities, we propose practical strategies such as enhancing resources at hospitals serving minority communities and improving access to comprehensive cancer care centers, as these have been identified as key contributors to disparities in minority-serving hospitals [29].

This study has several potential limitations that should be acknowledged. First, as a retrospective analysis using the CCR, the data are subject to the inherent constraints of registry-based research, including potential misclassification or incomplete capture of demographic and treatment details—for example, the CCR does not include granular data on the socioeconomic status of patients or detailed institutional-level information such as hospital volume. Furthermore, the categorization of multi-ethnic individuals based on primary reported ethnicity or grouping into ‘other/unknown’ may obscure associations specific to this population. While the CCR includes nearly all cancer cases in California, it may not capture nuanced clinical decision-making processes or reasons for treatment deviations. Understanding why a significant portion of patients’ treatment plans did not initially include surgery is essential. This “black box” of decision-making needs to be thoroughly investigated to determine the reasons surgery was not part of the initial plan for these patients.

Future research should investigate the underlying causes of the low rates of GCC in early-stage PDAC, explore the specific obstacles encountered by disadvantaged populations, and evaluate the effectiveness of interventions designed to enhance guideline adherence. This is particularly important given the lower guideline-concordance rates observed in pancreatic cancer compared to other malignancies like colon, lung, and breast cancer [30,31,32,33]. To improve outcomes for patients with PDAC, addressing the modifiable factors contributing to these disparities in care is crucial. Health policy initiatives should prioritize increasing healthcare providers’ awareness of treatment guidelines and ensuring the availability of adequate resources.

## 5. Conclusions

In conclusion, this study demonstrates that most patients with PDAC in California do not receive GCC, especially in the early stages, where adherence has the highest potential to impact survival. For stage 1 PDAC patients, low adherence to surgical guidelines represents a critical missed opportunity for curative intervention, further exacerbating poor survival rates for this already challenging cancer. We identified not undergoing surgery as the main driver of guideline non-concordance. Our findings also reveal notable disparities in care delivery associated with race, insurance status, and comorbidity burden, suggesting that socioeconomic factors and systemic inequities significantly affect treatment access and outcomes. Targeted interventions to improve guideline adherence in early-stage PDAC could profoundly impact survival rates and align care delivery more closely with national standards.

## Figures and Tables

**Figure 1 cancers-17-00170-f001:**
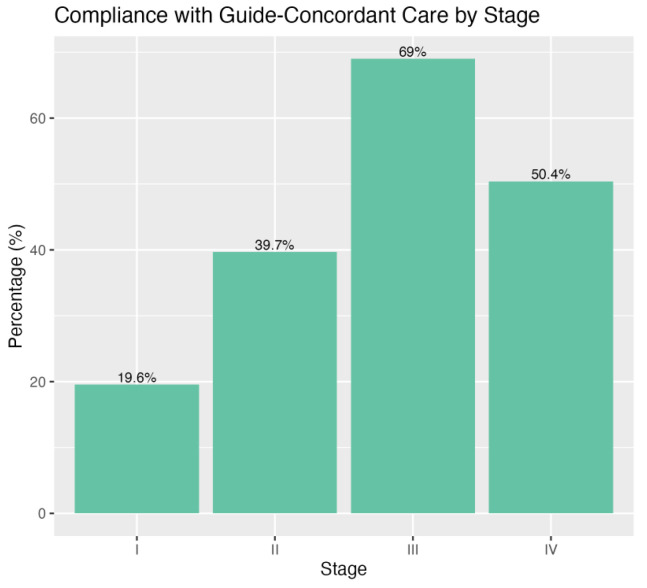
Percentage of patients receiving GCC by cancer stage. This bar graph displays the percentage of patients who received care in accordance with pancreatic cancer treatment guidelines in California, stratified by disease stage (I–IV). Stage III patients showed the highest compliance at 69%. In contrast, Stage I patients had the lowest at 19.6%.

**Figure 2 cancers-17-00170-f002:**
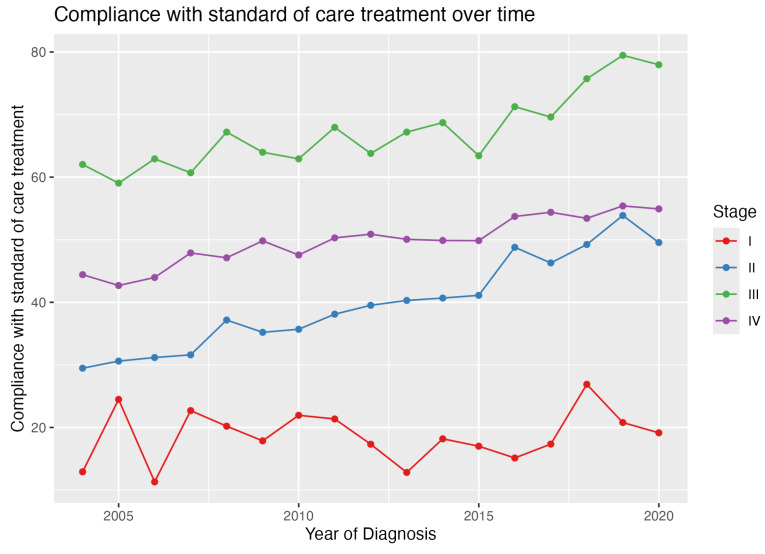
Trends in guide-concordant care over time, stratified by disease stage. This figure illustrates the changes in compliance with the standard of care treatment from 2004 to 2020, categorized by pancreatic cancer stages I–IV. Stage III patients consistently exhibited the highest rates of receiving GCC, with a slope of 1.017 and a significant *p*-value of <0.001. Conversely, Stage I patients had the lowest compliance rates, with a minimal slope increase of 0.123 and a non-significant *p*-value of 0.573. Stages II and IV also showed notable improvements, with slopes of 1.422 and 0.715 and *p*-values of <0.001, respectively.

**Figure 3 cancers-17-00170-f003:**
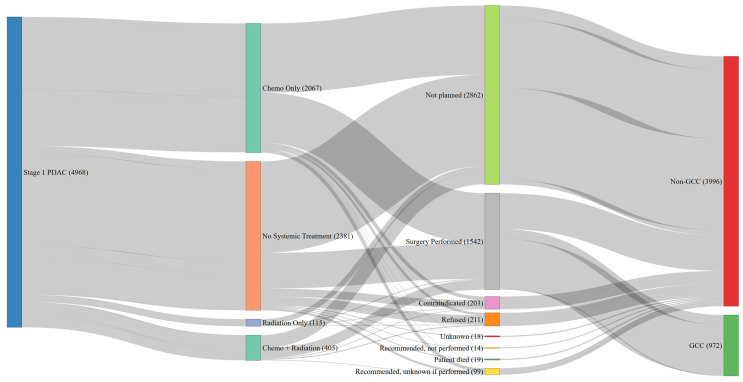
Sankey diagram of patient pathways for treatment and compliance in patients with stage 1 PDAC. This Sankey diagram Illustrates the treatment pathways and compliance outcomes for Stage 1 PDAC patients. Colors represent compliance categories (green for GCC and red for non-GCC). This figure highlights the role of lack of a plan for surgery contributing to non-GCC in stage 1 PDAC patients.

**Figure 4 cancers-17-00170-f004:**
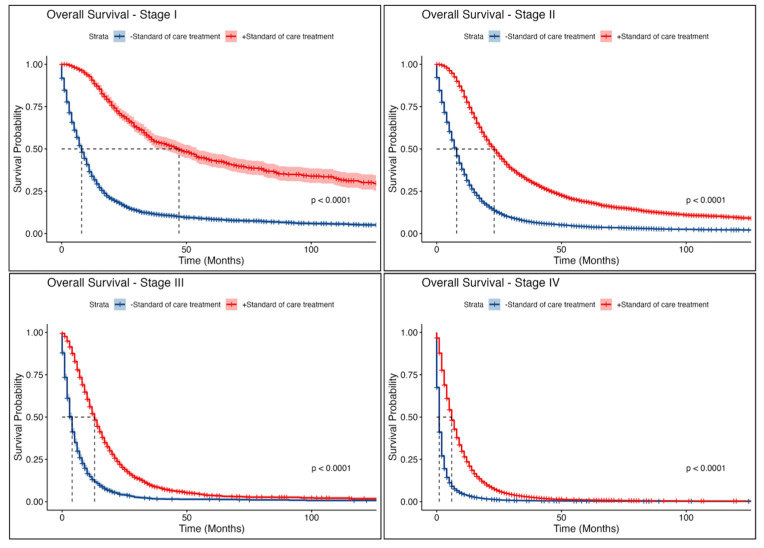
Stage-specific Kaplan–Meier survival curves by receipt of GCC.

**Table 1 cancers-17-00170-t001:** Demographic and clinical characteristics stratified by receipt of stage-specific GCC for pancreatic ductal adenocarcinoma in California.

Variable	GCC, N = 23,505 ^1^	No GCC, N = 26,841 ^1^	All Patients, N = 50,346 ^1^	*p*-Value ^2^
Age, years	66 (59, 74)	73 (64, 81)	70 (61, 78)	<0.001
Sex (Female)	11,044 (47%)	13,297 (50%)	24,341 (48%)	<0.001
Race				<0.001
White	14,379 (61%)	16,013 (60%)	30,392 (60%)	
Asian	2938 (12%)	3230 (12%)	6168 (12%)	
Black	1677 (7.1%)	2132 (7.9%)	3809 (7.6%)	
Hispanic	4339 (18%)	5263 (20%)	9602 (19%)	
Other	172 (0.7%)	203 (0.8%)	375 (0.7%)	
Treatment Center				<0.001
Non-NCI	38,377 (76%)	21,606 (80%)	16,771 (71%)	
NCI	11,969 (24%)	5235 (20%)	6734 (29%)	
Insurance				<0.001
Private	9806 (43%)	8507 (33%)	18,313 (37%)	
Medicare	11,781 (51%)	16,048 (62%)	27,829 (57%)	
No Insurance	319 (1.4%)	526 (2.0%)	845 (1.7%)	
Other	998 (4.4%)	986 (3.8%)	1984 (4.1%)	
Location of Residence				0.63
Metropolitan	22,019 (94%)	25,186 (94%)	47,205 (94%)	
Micropolitan	1010 (4.3%)	1094 (4.1%)	2104 (4.2%)	
Small Town	219 (0.9%)	256 (1.0%)	475 (0.9%)	
Rural Areas	255 (1.1%)	301 (1.1%)	556 (1.1%)	
CD Index				<0.001
0	8754 (42%)	795 (33%)	16,549 (37%)	
1	6226 (30%)	9592 (41%)	15,818 (36%)	
>2	5989 (29%)	6180 (26%)	12,169 (27%)	
Clinical Stage				<0.001
I	972 (4.1%)	3996 (15%)	4968 (9.9%)	
II	5000 (21%)	7598 (28%)	12,598 (25%)	
III	3760 (16%)	1690 (6.3%)	5450 (11%)	
IV	13,773 (59%)	13,557 (51%)	27,330 (54%)	

^1^ Median (IQR); n (%); ^2^ Wilcoxon Rank Sum Test; Pearson’s Chi-Squared Test; GCC, Guideline-Concordant Care; CD Index, Charlson–Deyo Comorbidity Index; NCI, National Cancer Institute-designated treatment center.

**Table 2 cancers-17-00170-t002:** Multivariable analysis of factors associated with receipt of stage-specific GCC.

	OR (95% CI)	*p*-Value
Age	0.95 (0.95–0.95)	<0.0001
Sex (Female)	0.98 (0.95–1.03)	0.46
Race		
White	—	
Asian	0.96 (0.90–1.02)	0.22
Black	0.74 (0.68–0.80)	<0.0001
Hispanic	0.78 (0.74–0.82)	<0.0001
Other	0.79 (0.62–1.01)	0.058
Insurance		
Private	—	
Medicare	0.95 (0.91–1.00)	0.042
No Insurance	0.40 (0.34–0.47)	<0.0001
Other	0.79 (0.71–0.88)	<0.0001
CD Index		
0	—	
1	0.93 (0.88–0.98)	0.0048
>2	0.68 (0.65–0.72)	<0.0001
Treatment Center		
Non-NCI	—	
NCI	1.64 (1.56–1.72)	<0.0001
Clinical Stage		
I	—	
II	2.30 (2.10–2.51)	<0.0001
III	8.49 (7.67–9.40)	<0.0001
IV	3.76 (3.47–4.09)	<0.0001
Location of Residence		
Metropolitan	—	
Micropolitan	0.97 (0.88–1.07)	0.55
Small Town	0.94 (0.76–1.15)	0.54
Rural Areas	0.94 (0.77–1.13)	0.51

CD Index, Charlson–Deyo Comorbidity Index; HR, Odds Ratio; CI, Confidence Interval; NCI, National Cancer Institute-designated treatment center.

**Table 3 cancers-17-00170-t003:** Cox proportional hazards analysis of factors associated with overall survival in pancreatic ductal adenocarcinoma patients.

	HR	95% CI	*p*-Value
Age	1.01	1.01–1.01	<0.001
Sex (Female)	0.95	0.94–0.97	<0.001
GCC			
No	—	—	
Yes	0.39	0.38–0.40	<0.001
Race			
White	—	—	
Asian	0.94	0.91–0.97	<0.001
Black	1.04	1.00–1.08	0.035
Hispanic	1.00	0.98–1.03	0.76
Other	0.99	0.88–1.11	0.84
Insurance			
Private	—	—	
Medicare	0.96	0.94–0.98	<0.001
No Insurance	1.16	1.07–1.26	<0.001
Other	1.04	0.99–1.10	0.11
Treatment Center			0.0001
Non-NCI	—	—	
NCI	0.80	0.78–0.82	
CD Index			
0	—	—	
1	1.09	1.06–1.11	<0.001
>2	1.33	1.30–1.36	<0.001
Clinical Stage			
I	—	—	
II	1.33	1.28–1.39	<0.001
III	2.41	2.29–2.52	<0.001
IV	4.65	4.47–4.83	<0.001
Location of Residence			
Metropolitan	—	—	
Micropolitan	1.03	0.98–1.08	0.30
Small Town	1.04	0.94–1.15	0.44
Rural Areas	1.01	0.92–1.10	0.90

GCC, Guideline-Concordant Care; CD Index, Charlson–Deyo Comorbidity Index; HR, Hazard Ratio; CI, Confidence Interval; NCI, National Cancer Institute-designated treatment center.

## Data Availability

The data for this research can be found at the CCR website by request. https://www.ccrcal.org/retrieve-data/data-for-researchers/how-to-request-ccr-data/ (accessed on 15 December 2024).

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
