# Peer review of "Ongoing Failure to Deliver Guideline-Concordant Care for Patients with Pancreatic Cancer"

_cancers, 2025, doi:10.3390/cancers17020170_

Round 1

Reviewer 1 Report

Comments and Suggestions for Authors Ongoing Failure to Deliver Guideline-Concordant Care for Patients with Pancreatic Cancer, is a very interesting research regarding treatment  and survival rate in patients with ductal pancreatic carcinoma. The paper is very important due to the the fact that  pancreatic cancer is   one of the deadliest cancers, with a very low general survival rate. The conclusion of the paper concerning patients in stage 1 that are low adherent to surgical treatment represents a red flag that should be highlighted. Also the localization of the  pancreatic tumor significantly influences its symptoms, treatment options, and prognosis, maybe authors could add this information in order to describe survival rate and prognosis.  Also could the authors specify if EUS FNA was used for the histopathological diagnostic? A very nice work and a large lot of patients included , which brings important highlights regarding this silent and severe type of cancer. 

Author Response

Comment 1: Also, the localization of the pancreatic tumor significantly influences its symptoms, treatment options, and prognosis; maybe authors could add this information in order to describe the survival rate and prognosis.  

Response 1: Thank you for your comment. We have categorized our cohort based on the AJCC staging framework that classifies tumors based on size, nodal involvement, and metastatic status. We have clarified this point in the revised manuscript and emphasized the role of standardized staging in guiding treatment decisions and survival analysis. Thank you for raising this important aspect.

Comment 2: Also could the authors specify if EUS FNA was used for the histopathological diagnostic? A very nice work and a large lot of patients included which brings important highlights regarding this silent and severe type of cancer.

Response 2: Thank you for your question. While EUS-FNA is a standard technique for obtaining tissue samples for histopathological diagnosis of pancreatic ductal adenocarcinoma, the California Cancer Registry data used for this study does not provide specific details on the diagnostic methods employed, including the use of EUS-FNA. We have added this limitation to the revised manuscript to ensure transparency. 

Thank you for your thoughtful feedback, which has helped improve our manuscript; we hope these revisions meet your expectations and welcome any further suggestions.

Reviewer 2 Report

Comments and Suggestions for Authors

Dear Author, I give you the following comment. Please address this in your manuscript to enhance the readability and understanding of your manuscript.

Major Comments

  1. Regarding Data Analysis
    • Could the authors elaborate on how the multivariable logistic regression model was validated, particularly for its robustness in identifying factors influencing the receipt of guideline-concordant care (GCC)?
  2. Concerning Study Population
    • The study primarily uses data from California. How generalizable are these findings to other regions or populations, and are there demographic or healthcare system differences that could influence the outcomes?
  3. About Barriers to Care
    • The study identifies racial and insurance-related disparities in receiving GCC. Could the authors expand on actionable strategies or interventions that might effectively address these disparities?
  4. Details on Survival Benefits
    • The survival benefit of adhering to GCC is significant, particularly in stage 1 PDAC. Could the authors clarify whether any specific component of GCC (e.g., surgery, chemotherapy) contributed more heavily to this survival advantage?
  5. Discussion on Limitations
    • While the study acknowledges disparities, are there any limitations related to the registry data (e.g., missing variables or potential biases) that might impact the validity of the findings?

Minor Comments

  1. On Definitions
    • Could the authors provide more detail in the main text or supplement on how the National Comprehensive Cancer Network guidelines were operationalized into the definition of GCC?
  2. Clarity in Results Section
    • In the Results section, the percentages for GCC by stage are provided. Could these figures be visually represented (e.g., in a bar chart) for clearer communication?
  3. Ethnic Group Categorization
    • How were multi-ethnic individuals categorized in the analysis, and could this approach influence the reported associations?
  4. Insurance Classification
    • The manuscript mentions "no insurance" as a factor. Were different types of insurance (e.g., private vs. public) analyzed separately, and if so, could those findings be included?
  5. On Statistical Reporting
    • Some odds ratios (ORs) and hazard ratios (HRs) are reported without confidence intervals. Could these be added for greater statistical transparency?

These questions aim to address both overarching concerns and specific technical details that could impact the robustness and clarity of the study's findings.

Best Regards

Comments on the Quality of English Language

Fine

Author Response

Comment 1: Concerning Study Population 
The study primarily uses data from California. How generalizable are these findings to other regions or populations, and are there demographic or healthcare system differences that could influence the outcomes? 

Response 1: Thank you for this thoughtful question. While our study primarily uses data from the California Cancer Registry (CCR), which reflects the unique characteristics of California, we believe the state’s diverse demographic distribution closely approximates the overall makeup of the United States better than many prior studies. California's population includes a significant proportion of racial and ethnic minorities, immigrant groups, and individuals across a wide range of socioeconomic statuses, providing a broader representation of the U.S. population than data from more homogenous regions. 
We also recognize that regional differences in healthcare practices and systems may impact GCC. For example, Salami et al. (reference) found that patients in the Northeast were more likely to be recommended for surgery and had better pancreatic cancer outcomes compared to other regions, while patients in the West (including California) and Southeast were less likely to receive surgical recommendations. Similarly, Kasumova et al. (reference) reported that patients in the Northeast had the best outcomes, particularly for stage I pancreatic cancer, with significantly higher odds of receiving multimodality therapy after resection compared to other regions. Our findings reveal significant racial and insurance-related disparities in the receipt of guideline-concordant care. Notably, Black and Hispanic patients were less likely to receive GCC, despite comparable rates of surgical refusal to White patients. These disparities, along with lower GCC rates among uninsured patients, underscore the urgent need for targeted strategies to address inequities in cancer care access and delivery. Thank you for the suggestion. We have underscored these disparities and addressed aspects of regional disparities in the revised Discussion section.

Comment 1: About Barriers to Care 
The study identifies racial and insurance-related disparities in receiving GCC. Could the authors expand on actionable strategies or interventions that might effectively address these disparities? 

Response 1: Thank you for this thoughtful question. Addressing disparities in receiving guideline-concordant care (GCC) requires strategies that account for both systemic barriers and patient-specific challenges. Our findings align with observations by Lima et al., who noted that minority-serving hospitals (MSH) have lower rates of GCC compared to non-minority-serving hospitals. Lima et al. also pointed out that multidisciplinary and multimodal PDAC care requires frequent visits, close surveillance, and follow-up adherence, which can be hindered by longer distances to non-minority-serving hospitals. This barrier is particularly relevant to our study, where we found that patients with stage I PDAC from small towns had a hazard ratio of 1.52, indicating a significantly higher risk of adverse outcomes. To mitigate these disparities, we propose the following actionable strategies: enhancing resources at Minority-Serving Hospitals and improving accessibility to Multidisciplinary Care Centers. This was added to our discussion. Additionally, a new supplementary table has been included to provide further detail on these findings and their implications.

Comment 2: Details on Survival Benefits 
The survival benefit of adhering to GCC is significant, particularly in stage 1 PDAC. Could the authors clarify whether any specific component of GCC (e.g., surgery, chemotherapy) contributed more heavily to this survival advantage? 

Response 2: Thank you for this insightful question. To address this, we have added a supplementary table presenting the Cox proportional hazard ratios for stage 1 PDAC patients who received surgery, chemotherapy, or radiation therapy individually. Our analysis shows that patients who underwent surgery experienced the greatest reduction in mortality compared to those who did not receive surgical intervention.

Comment 3: Discussion on Limitations 
While the study acknowledges disparities, are there any limitations related to the registry data (e.g., missing variables or potential biases) that might impact the validity of the findings?

Response 3: Thank you for raising this important point. Limitations related to the registry data include missing variables such as granular data on socioeconomic status and hospital volume, both of which are important factors that could influence adherence to guideline-concordant care and patient outcomes. These limitations have been acknowledged and added to the revised Limitations section of the Discussion to ensure transparency. 

Comment 4: Minor Comments On Definitions 
Could the authors provide more detail in the main text or supplement on how the National Comprehensive Cancer Network guidelines were operationalized into the definition of GCC? 

Response 4: Thank you for this insightful question. A more detailed explanation of the operationalization process has been added to the Methods section and a supplementary table provides further clarification.

Comment 5: Clarity in Results Section 
In the Results section, the percentages for GCC by stage are provided. Could these figures be visually represented (e.g., in a bar chart) for clearer communication? 

Response 5:
Thank you for your suggestion. We believe this figure effectively communicates the stage-specific differences in GCC adherence. The percentages for guideline-concordant care by stage are represented visually in Figure 1. 

Comment 6:Ethnic Group Categorization 
How were multi-ethnic individuals categorized in the analysis, and could this approach influence the reported associations? 

Response 6: Thank you for raising this important question. In our analysis, individuals identifying as multi-ethnic were categorized based on the primary ethnicity reported in the registry data. If no primary ethnicity was indicated, they were grouped into an “other/unknown” category. While this approach ensures the inclusion of all individuals, we acknowledge that it may dilute or obscure associations specific to multi-ethnic populations due to the heterogeneity within these categories. This potential limitation has been acknowledged in the revised Discussion section.

Comment 7: Insurance Classification 
The manuscript mentions "no insurance" as a factor. Were different types of insurance (e.g., private vs. public) analyzed separately, and if so, could those findings be included? 

Response 7: Thank you for your question. The analysis of different types of insurance, including Medicare, private insurance, and other categories, is included in Tables 1-3 of the manuscript.

Comment 8 On Statistical Reporting 
Some odds ratios (ORs) and hazard ratios (HRs) are reported without confidence intervals. Could these be added for greater statistical transparency? 

Response 8: Thank you for pointing this out. We agree that including confidence intervals (CIs) for odds ratios (ORs) and hazard ratios (HRs) is essential for statistical transparency and interpretation. Confidence intervals have been added to all reported ORs and HRs in the Results section, as well as to the corresponding tables and figures where applicable. 

Thank you for your thoughtful feedback, which has helped improve our manuscript; we hope these revisions meet your expectations and welcome any further suggestions.

Reviewer 3 Report

Comments and Suggestions for Authors

This paper analyzes background factors for adherence to guideline-concordant care (GCC) in pancreatic cancer and their relationship to prognosis.

 These are very interesting results, examined from a large data set of 50,000 patients in California. Please answer questions and comments

 The background factors for adherence to GCC are being analyzed, but factors such as hospital volume and factors in cancer centers and general hospitals should be added to the analysis.

 Discussion section, please specify the 'nihilistic attitudes' of the health care provider

 Is the adherence rate of GCC in this study lower than for colon, lung, or breast cancer etc. ?  If it is low, what are the reasons for this low rate?

 What are the key points of revision of the NCCN guidelines to improve the GCC adherence rate?

In the text

Line 118, add the year to each GCC rate.

Line 159, isn't the improvement rate 6%?

Please insert the reference number.

Author Response

Comment 1: The background factors for adherence to GCC are being analyzed, but factors such as hospital volume and factors in cancer centers and general hospitals should be added to the analysis. 

Response 1: Thank you for this valuable suggestion. We agree that institutional factors such as hospital volume and the distinction between cancer centers and general hospitals may play a critical role in adherence to guideline-concordant care (GCC). Thus, we have added NCI designation into our analysis. Unfortunately, our current dataset does not include detailed institutional-level information such as hospital volume, which limits our ability to directly analyze these factors. We have updated our manuscript to reflect this limitation.

Comment 2: Discussion section, please specify the 'nihilistic attitudes' of the healthcare provider
Response 2: Thank you for highlighting this point. By "nihilistic attitudes," we refer to the misperceptions among some healthcare providers regarding the outcomes of pancreatic cancer treatment. For instance, Woodmass et al. found that nearly one-third of family physicians overestimated perioperative mortality to exceed 10%, despite evidence showing rates of 1-4% in high-volume centers. Additionally, only 41% of family physicians recognized surgery as potentially curative, compared to 77% of surgeons. These misperceptions likely limit referrals for surgical resection, reducing patient access to guideline-concordant care (GCC). We have clarified this in the revised Discussion section and emphasized the need for provider education to address these misconceptions.

Comment 3: Is the adherence rate of GCC in this study lower than for colon, lung, or breast cancer, etc.?  If it is low, what are the reasons for this low rate? What are the key points of revision of the NCCN guidelines to improve the GCC adherence rate? 

Response 3: Thank you for your question. Adherence to GCC in PDAC is lower compared to other cancers. For example, 76% of metastatic non-small cell lung cancer patients receive first-line guideline therapy, and complete guideline concordance in colon cancer is 24.7%, with major non-concordance often due to missing chemotherapy. Breast cancer adherence rates vary but are influenced by access to care, race, and age, with older and nonwhite patients less likely to receive guideline care.
In PDAC, lower adherence stems from its aggressive progression, limited surgical eligibility, provider misperceptions, and logistical barriers like access to high-volume centers. Revisions to NCCN guidelines could address these issues by emphasizing multidisciplinary care, reducing delays in treatment, standardizing the management of borderline resectable cases, and implementing telemedicine and navigation programs to improve access and follow-up adherence. These points have been incorporated into the revised Discussion section to address your feedback. 

Comment 4: In the text 
Line 118, add the year to each GCC rate. 
Response: Thank you for this suggestion. We have clarified the year for each GCC rate.
Line 159, isn't the improvement rate 6%? 
Response: Thank you for your question. That is correct.
Please insert the reference number. 

Thank you for pointing this out. The appropriate reference number(s) have been inserted in the relevant sections of the manuscript to ensure clarity and proper citation.

Thank you for your thoughtful feedback, which has helped improve our manuscript; we hope these revisions meet your expectations and welcome any further suggestions.

Reviewer 4 Report

Comments and Suggestions for Authors

The article " Ongoing Failure to Deliver Guideline-Concordant Care for Patients with Pancreatic Cancer" characterizes treatment patterns and assess factors associated with receiving Guideline-Concordant Care (GCC) among patients with pancreatic ductal adenocarcinoma (PDAC) in California. GCC is important to be considered to ensure that all patients receive the highest quality of care which will then reflect to produce the best possible chance of achieving a favorable outcome and/or quality of life. This study seems upgraded latest version of previous similar articles covering larger patient cohort. This study highlights a potential gap between evidence-based guidelines and the actual care received by patients which is a challenge in the treatment of pancreatic cancer and can/need to be addressed for better management of the disease. Factors behind the potential gap need to be identified/addressed by caregivers.

The article is well-written, analyzed, discussed, and presented. The article on itself is technically fit for the publication. However, some minor revisions suggested are:

1.    Please introduce GCC for PDAC in more detail in introduction, so that readers are familiar before going into the results/discussion/conclusion of the study.

2.    In methods section please update, if any update and/or changes in GCC has occurred for the study period (2004-2020), and how was that considered for the result analysis.

3.    There is more scope to discuss the current gap and how that could be rectified. If example of such gaps existing in other diseases and/or how those were rectified can be added.

Author Response

Comment 1: Please introduce GCC for PDAC in more detail in the introduction so that readers are familiar with it before going into the results/discussion/conclusion of the study. 
Response 1: Thank you for your suggestion. A more detailed introduction to guideline-concordant care for pancreatic ductal adenocarcinoma, including its core components, has been included in the Introduction section.

Comment 2: In the methods section, please update if any updates and/or changes in GCC have occurred for the study period (2004-2020) and how was that considered for the result analysis. 
Response 2: Thank you for this question. While the NCCN guidelines have evolved over time to incorporate advancements in research and clinical practice, the general stage-based approach to the management of pancreatic ductal adenocarcinoma (PDAC) has remained consistent throughout the study period (2004–2020). These updates primarily focused on refinements in treatment strategies, such as the expanded role of neoadjuvant therapies and clarification of borderline resectable disease. (Tempero et al., 2012, 2017, 2021) 

Given this consistency in the overarching framework, the analyses in our study were conducted under the assumption that the stage-based treatment principles remained comparable across the study period. Any potential variations due to guideline updates were minimized by assessing adherence to core recommendations applicable across all guideline versions. This has been clarified in the revised Methods section.

Comment 3: There is more scope to discuss the current gap and how that could be rectified. If example of such gaps existing in other diseases and/or how those were rectified can be added.

Response 3: Thank you for the suggestion. We have addressed this by discussing why GCC rates for PDAC are lower compared to cancers like colon, breast, and lung, expanded on the potential role of nihilistic attitudes among providers, highlighted regional disparities between the West and other U.S. regions, and touched on potential interventions such as enhancing resources at minority-serving hospitals and improving accessibility to multidisciplinary care centers to address demographic disparities. These points have been added to the Discussion section.

Thank you for your thoughtful feedback, which has helped improve our manuscript; we hope these revisions meet your expectations and welcome any further suggestions.